genetics/bioengineering

poplar, transgenic seedlings, *ARK1* gene, transcriptome profile, real-time quantitative polymerase chain reaction (RT-qPCR) analysis

**Author for correspondence:**
Hanyao Zhang
e-mail: zhanghanyao@hotmail.com

# Enhancing the expression of *ARK1* genes in poplar leads to multiple branches and transcriptomic changes

Xiaozhen Liu[1], Zhiming Zhang[2], Wen Bian[2], Anan Duan[1] and Hanyao Zhang[1]

[1]Key Laboratory for Forest Resources Conservation and Utilization in the Southwest Mountains of China, Southwest Forestry University, Ministry of Education, Kunming, Yunnan 650224, People's Republic of China
[2]Key Laboratory of Biodiversity Conservation in Southwest China, State Forest Administration, Southwest Forestry University, Kunming, Yunnan 650224, People's Republic of China

HZ, 0000-0001-7440-4348

The *ARBORKNOX1* (*ARK1*) gene is an important gene for regulating plant growth and development; however, transcriptomic responses of enhancing expression of *ARK1* gene in poplar are poorly investigated. To provide insight into the gene function of the *ARK1* gene in poplar, the *ARK1* transgenic poplar '717' and '84 K' lines were obtained, the morphology of transgenic plants was observed, and transcriptome profiles were compared. The results showed that there were multiple branches in *ARK1* transgenic seedlings compared with non-transgenic seedlings. The results of transcriptome analysis showed that there were significant differences in transcriptome profiles between the transgenic lines of '717' and '84 K', and between non-transgenic lines (CK) and transgenic plants. The real-time quantitative polymerase chain reaction (RT-qPCR) analysis confirmed the expression levels of the genes involved in the pathway of zeatin biosynthesis and brassinosteroid biosynthesis. The increase in expression levels of *AHP* and *CYCD3* was related to multiple branches. Enhancing the expression of the *ARK1* gene in poplar seedlings leads to multiple branches and transcriptomic changes.

## 1. Introduction

Poplar (*Populus*) is widely used in ecological protection, urban greening, industry and fibre production because of its rapid growth, wide adaptability and wide distribution [1]. The whole-genome sequencing of some species of *Populus* has been

completed, and it is an ideal model plant for forest genetic breeding and improvement [2,3]. Poplar is the natural host of *Agrobacterium tumefaciens*, which is convenient for plant genetic transformation using the *A. tumefaciens*-mediated method [4]. Besides, poplar is the fastest growing tree species in the temperate zone, with a variety of agronomic characters and advantages [5]. Poplar and its hybrids are considered to be the preferred perennials for the production of bioenergy raw materials [5].

*ARK1* is a key transcription factor that regulates the activity of cambium formation and cell differentiation [6]. It belongs to the *KNOX* gene family (class I *KNOX*, a subfamily of *KNOTTED1*-like homeobox genes). The *KNOX* gene family encodes homeobox proteins, which play an important regulatory role in plant growth and development [6]. *KNOX* Class I subfamily genes are mainly expressed in the plant meristem and are the key factors necessary for meristem genesis and maintenance [7]. By regulating cell differentiation that related to organogenesis, it finally affects the morphogenesis of lateral organs [6].

The *ARK1* gene is known to be important for plant growth and development. Zeatin and brassinosteroids were found to be related to the branching of plants [8,9], and the growth of plant phenotype is affected by phenylpropanoid [10]. In recent years, the shoot development and formation of rice were found to be related to zeatin [8]. Lv *et al*. [11] found that zeatin is related to the 'shoot branching' of *Cremastra appendiculata*. Han *et al*. [12] also found that the zeatin signalling pathway is related to multiple branches in *Betula platyphylla* × *Betula pendula*. Brassinosteroids play important roles in the regulation of shoot branching [9]. That brassinosteroid transcriptional effector *BES1* regulates shoot branching was found by Wang *et al*. [13]. Phenylpropanoid pathway was found to be significantly related to plant development [14]. Merali *et al*. [15] found that phenylpropanoid-metabolizing enzyme in tobacco reveals essential roles of phenolic precursors in normal leaf development and growth. Lignin is a phenylpropanoid-derived heteropolymer important for the strength and rigidity of the plant secondary cell wall [16]. Hence, the researches on zeatin and brassinosteroids signalling pathways and phenylpropanoid biosynthesis pathways should be focused on, when the *ARK1* gene is transformed and transcriptome analysis is conducted in plants.

Hybrid poplar '717' (*Populus tremula* × *P. alba*) and '84 K' (*P. alba* × *P. tremula* var. *glandulosa*) are excellent lines of poplar interspecific hybrids with strong stress tolerance and are of great application value for vegetation restoration, prevention of soil erosion and saline-alkali land restoration, and can also be used for the development of biomass energy and fibre energy [3,17,18]. Hybrid poplar '717' and '84 K' are perennials, compared with annual herbs such as rice (*Oryza sativa*) and *Arabidopsis thaliana*; they have unique advantages for studying the wood formation and seasonal dormancy. Poplar produces a large number of repeated sequences in its evolutionary process, and its regulation mechanism is more complex, so it is particularly important to use '717' and '84 K' poplar as research objects for gene function analysis.

The growth of plants, especially that of wood plants, is still very slow and is difficult to meet the needs of industrial development for wood, biomass, paper, fuel and biomaterials [19,20]. The *ARK1* gene has been found to be an important gene for regulating plant growth and development in poplar in previous studies [6,7]. To investigate the function of the *ARK1* gene may benefit the plant growth to meet the needs. In this study, to analyse the function of the *ARK1* gene, the poplar lines '717' and '84 K' were subjected to gene transform, the morphology and transcriptome profiles were compared between transgenic lines and control plants, and the expression levels of some genes were verified by real-time quantitative polymerase chain reaction (RT-qPCR).

# 2. Methods

## 2.1. Plant materials

In this study, leaves of hybrid poplar '717' and '84 K' from the plant nursery of the Southwest Forestry University were used for gene transformation. Hybrid poplar '717' and '84 K' were introduced from Northwest Agriculture and Forestry University in March 2015.

## 2.2. Vector construction

Plant DNA was extracted from a leaf of '84 K' using the cetyl trimethylammonium bromide (CTAB) method [21]. The primer sequences for the amplification of the *ARK1* gene were ARBF: 5′-GGAGAGGACACGCTC-GAGATGGAGGGTGGTGATGGTG-3′ and ARBR: 5′-ATCCTTGTAGTCGAATTCAAGCAGTGTGGGA GAGATGTC-3′. After the objective gene was amplified, a QIAquick PCR purification kit (Qiagen Inc., Valencia, CA) was used to retrieve the target gene band. Then, the target gene was digested with *Xho*I and

*Eco*RI and ligated to large fragments of PART-CAM-FLAG digested with *Xho*I and *Eco*RI to construct the vector (see electronic supplementary material, figure S1). The ligated recombinant fragments were transformed into DH5α competent cells and screened using polymerase chain reaction (PCR) technology; the PCR primers were the same as those used in the section of molecular detection of transgenic plants. Positive colonies were sequenced, and constructed vectors and related strains were obtained successfully.

## 2.3. Vector transformation of Agrobacterium tumefaciens

The plasmid was extracted from DH5α containing an *ARK1* gene vector and transformed into the *A. tumefaciens* strain LBA4404, and PCR was used to select positive strains.

## 2.4. Gene transformation and detection of '717' and '84 K'

The leaves of '717' and '84 K' poplar infected with *A. tumefaciens* were inoculated using a callus induction medium (Murashige and Skoog (MS) + naphthalene acetic acid (NAA) 1.0 mg $l^{-1}$ + 6-benzyl aminopurine (6-BA) 1.0 mg $l^{-1}$) and co-cultured at 28°C for 2 days. The co-cultured calli were then washed with sterile water approximately three times, dried with aseptic paper, and transferred to the sterile differentiation medium (MS + 6-BA 1.0 mg $l^{-1}$ + zeatin (ZT) 0.4 mg $l^{-1}$) with selective pressure (200 mg $l^{-1}$ carbenicillin + 200 mg $l^{-1}$ kanamycin). The calli were selectively cultured under a light cycle of 16/8 h at 28°C. After approximately 28 days, when the adventitious buds grew to 2–3 cm, they were transferred to the rooting medium (MS + NAA 0.02 mg $l^{-1}$ + indole-3-butyric acid (IBA) 0.6 mg $l^{-1}$) containing selective pressure (200 mg $l^{-1}$ carbenicillin + 200 mg $l^{-1}$ kanamycin) for rooting culture. After the adventitious roots were produced, the plants were transplanted to basins in the greenhouse.

## 2.5. Molecular detection of transgenic plants

DNA was extracted by using the CTAB method from seedlings as described by Ye *et al.* [22]. The corresponding specific detection primers were designed according to the constructed expression vector. The upstream primer was the internal sequence of the *ARK1* target gene, and the downstream primer was the sequence of *NPT II*. The primer sequences were 5′-AAGATCCAGCCCTTGACCAA-3′ (forward) and 5′-CATTGCCATCACCACAACCA-3′ (reverse). The PCR amplification procedure was as follows: 98°C, 3 min; 35 cycles: 94°C, 30 s, 55°C, 30 s, 72°C, 5 min; 72°C, 10 min. After the PCR reaction was carried out, the PCR products were detected by 1.2% agarose gel electrophoresis (with 0.5 μg mol$^{-1}$ ethidium bromide).

*Bam*HI~*Xho*I digested fragments of the *ARK1* gene were $^{32}$P-labelled, and Southern blot analysis was conducted according to the method reported by Liu *et al.* [23].

## 2.6. Detection of morphological changes in transgenic plants

After being cultured for approximately 45 days, the morphological differences of tissue culture seedlings between transgenic and non-transgenic poplars of '717' and '84 K' hybrid poplars were compared.

## 2.7. Transcriptome sequencing

After 45 days of culture, the stems were harvested, and the mRNA of transgenic and non-transgenic poplar samples of '717' and '84 K' poplar was extracted using a Qiagen RNeasy Mini Kit (Qiagen Inc., Valencia, CA) and inversely transcribed into cDNA (using the SuperScript™ VILO™ cDNA Synthesis Kit, Roche Diagnostics GmbH, Mannheim, Germany). The cDNA of transgenic '717' and '84 K' hybrid poplar and non-transgenic '717' and '84 K' hybrid poplar was analysed using the sequencing platform Illumina HiSeqTM 2500. Pathway analysis was performed using DEseq2 software [24]. The differentially expressed genes (DEGs) were screened with padj <0.05 and abs(log2FoldChange) >1. The genome of '717' hybrid poplar was downloaded from http://aspendb. uga.edu/index.php/databases/spta-717-genome to be the reference genome.

## 2.8. Hormone content determination

After 45 days of culture, three stems of transgenic '717' and '84 K' poplar and three stems of non-transgenic '717' and '84 K' poplar were harvested. The contents of zeatin and brassinosteroids were determined according to the method reported by Li *et al.* [25].

**Table 1.** Primer sequences of target genes (TG) and reference gene (RG) used in RT-qPCR.

| gene name | primer | sequence (5'-3') | length (bp) |
|---|---|---|---|
| elongation factor $EF1\beta$ (RG) | forward | GACAAGAAGGCAGCGGAGGAGAG | 269 |
| | reverse | CAATGAGGGAATCCACTGACACAAG | |
| $ARK1$ | forward | CATCCATCACCACAAACTGC | 165 |
| | reverse | ATTGGTGGAGCAGGCATTAC | |
| $HAP_{-84\,K}$ | forward | CCCAGTTGATCGTCTCGTCA | 221 |
| | reverse | TTGGCTTGCTTCAACATCCC | |
| $HAP_{-717}$ | forward | ATCAGCTCATTGGAAGCAGC | 197 |
| | reverse | CAGCTGCCAGTACTCTCAGT | |
| $CYCD3_{-84\,K}$ | forward | TAGAACCCAGTCTTGCAGCA | 240 |
| | reverse | GAAGACACGGATGATGCCAC | |
| $CYCD3_{-717}$ | forward | TGGAGGTGTGAGCGTTTACT | 226 |
| | reverse | CGTTGGTTTTGACTGCCTGA | |
| $PER\,27_{-84\,K}$ | forward | CTTTGTGAGGGGCTGTGATG | 175 |
| | reverse | CCACAATGGCCATGATGTCC | |
| $PER\,27_{-717}$ | forward | CCCTGCATTTTACGACAGCT | 161 |
| | reverse | TCGTCGAGTAAGATGGAGGC | |

## 2.9. RT-qPCR verification

After 45 days of culture, the stems were harvested and the mRNA of three transgenic and three non-transgenic '717' and '84 K' poplars was extracted and reverse transcribed into cDNA. We verified the expression of *ARK1*, *AHP*, *CYCD3* and *PER27* genes in the zeatin biosynthesis pathway, brassinosteroid biosynthesis and the phenylpropanoid biosynthesis pathway. The RT-qPCR analysis was conducted according to a previous report [26]. $EF1\beta$ was used as the housekeeping gene for normalizing, and the $2^{(-\Delta\Delta\,Ct)}$ method [26] was used to analyse the expression data. The primers used for RT-qPCR analysis are shown in table 1.

# 3. Result

## 3.1. Transformants of *ARK1*

The DNA of transgenic '717' and '84 K' seedlings with the *ARK1* gene was detected by PCR. Eleven out of 35 '717' seedlings and 14 out of 41 '84 K' seedlings with a transformation marker were identified, and a total of 32.22% plants were found to be PCR positive. It illustrated that the genetic transformation of '717' and '84 K' poplar with *ARK1* overexpression vector was successful. PCR positive '717' and '84 K' transgenic lines were selected for Southern blot, and all of them were positive. Part of the Southern blot results is shown in electronic supplementary material, figure S2. The promotor for enhancing *ARK1* gene is *CaMV 35S*. After RT-qPCR analysis, the expression levels of eight transgenic '717' lines and ten '84 K' lines were found to be at least twice more than that of the control seedlings. The results of the RT-qPCR analysis showed that the average expression levels of the *ARK1* gene in transgenic '717' and '84 K' lines were approximately 2.95 times (using $EF1\beta$ as the reference gene, see [22]) and 3.16 times that of the control seedlings ($p$-value = 0.000028 and 0.000019, significant at $p < 0.01$, see electronic supplementary material, figure S3).

## 3.2. Phenotypic changes of transgenic poplars

There were significant differences in appearance between transgenic '717' and '84 K' poplar seedlings compared with non-transgenic seedlings. The stem segment of transgenic poplar '717' and '84 K' was slender, usually fascicled, and had multiple branches without a slender petiole or slender leaf shape,

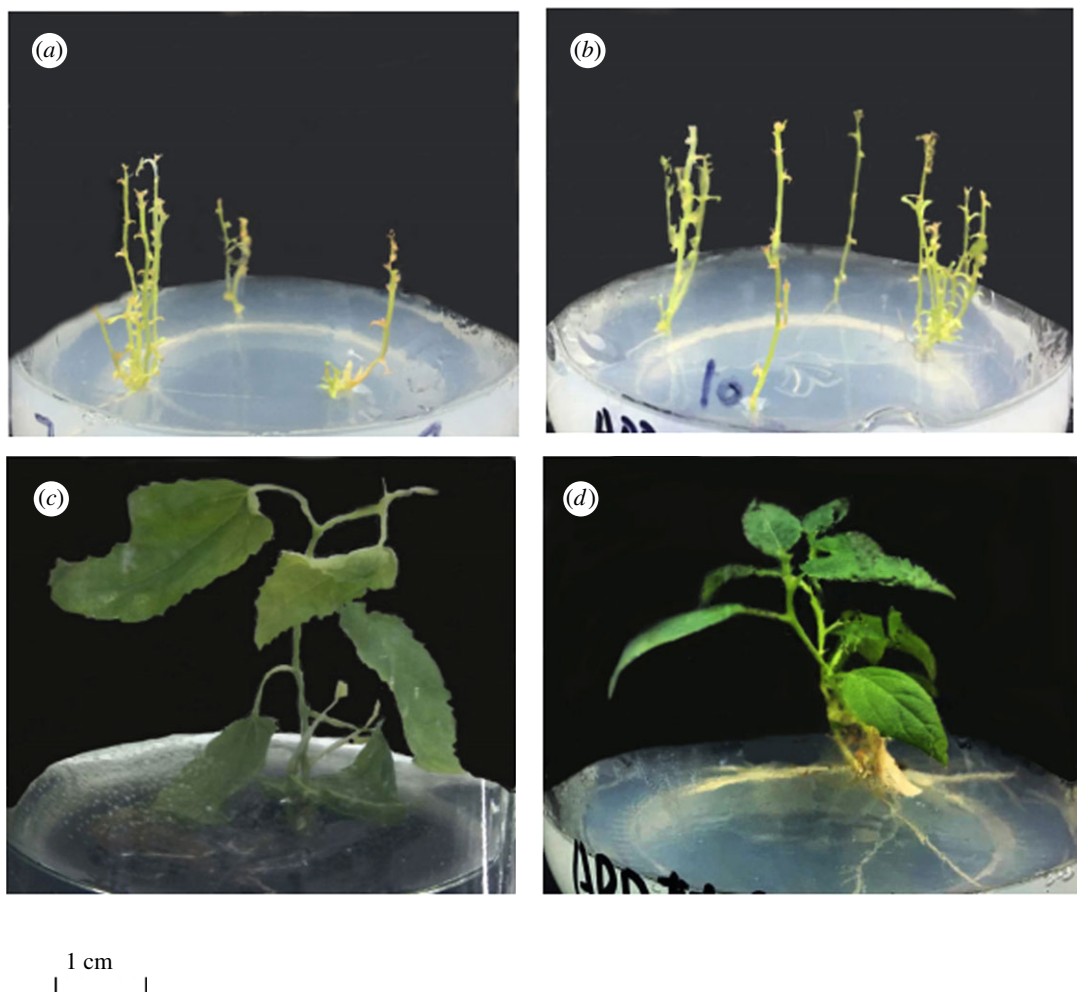

1 cm

**Figure 1.** Phenotype of transgenic plants. (*a*) Transgenic '717'; (*b*) transgenic '84 K'; (*c*) none-transgenic '717'; (*d*) none-transgenic '84 K'. These seedlings were cultured for approximately 45 days.

and some leaves were not fully developed and curling (as shown in figure 1). After being cultured for approximately 45 days, the branches in all PCR positive '717' and '84 K' lines were approximately 4.73 times and 5.24 times that of control seedlings (*p*-value = 0.00516 and 0.00419, significant at $p < 0.01$, see electronic supplementary material, figure S4). After half a year of culture, the leaves changed back to their original state, but the situation of multiple branching remained unchanged.

## 3.3. Transcriptome profile changes of transgenic poplars

The original data were filtered by quality control, and the results showed that the sequencing quality was good and met the requirements of building the database (see electronic supplementary material, table S1). Six hundred and forty-one DEGs were screened in the *ARK1* transgenic poplar '717', including 389 upregulated genes and 252 downregulated genes (see [22]). Seven hundred and thirty-six DEGs were screened from *ARK1* transgenic poplar '84 K', including 382 upregulated genes and 354 downregulated genes. Between transgenic poplar '717' and control plants, DEGs were enriched in 25 gene ontology identity document (GOIDs); these GOIDs were described as polysaccharide metabolic process, cellular glucan metabolic process, glucan metabolic process, response to oxidative stress, aminoglycan metabolic process, aminoglycan catabolic process, chitin metabolic process and so on. Upregulated DEGs were mainly involved in hormone (including zeatin and brassinosteroids) metabolic process, translational termination, protein complex disassembly, cellular protein complex disassembly and macromolecular complex disassembly. Downregulated DEGs were mainly involved in phenylpropanoid metabolic process, aminoglycan metabolic process, chitin catabolic process, amino sugar metabolic process, cell wall macromolecule catabolic process, cell wall macromolecule metabolic

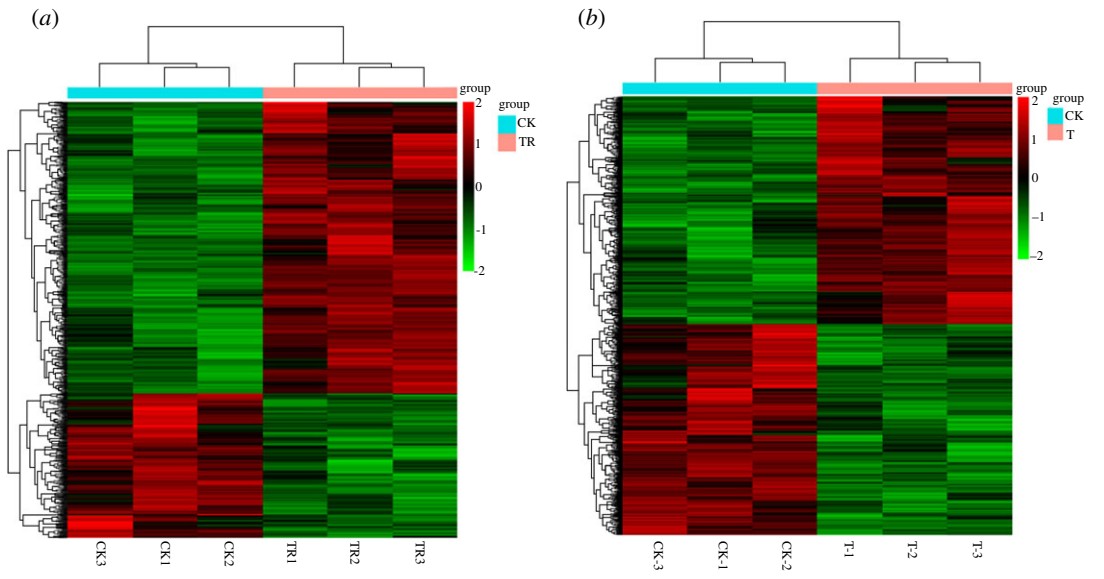

**Figure 2.** clustering heatmap of differentially expressed genes. (*a*) Transgenic '717' lines and control plants; (*b*) transgenic '84 K' lines and control plants; CK, 1–3, '717'; CK_1-3, '84 K' lines; TR1-3, transgenic '717' lines; T_1-3, transgenic '84 K' lines.

process, amino sugar catabolic process, glucosamine-containing compound metabolic process and glucosamine-containing compound catabolic process. Between transgenic poplar '84 K' and control plants, DEGs were enriched in 29 GOIDs, these GOIDs were described as cellular polysaccharide metabolic process and monocarboxylic acid metabolic process, response to oxidative stress, aminoglycan metabolic process, aminoglycan catabolic process, chitin metabolic process, and so on. Up-regulated DEGs were mainly involved in hormone (including zeatin and brassinosteroids) metabolic process, translational termination, cellular protein complex disassembly, cellular component disassembly and macromolecular complex disassembly. Downregulated DEGs were mainly involved in phenylpropanoid metabolic process, aminoglycan metabolic process, aminoglycan catabolic process, chitin metabolic process, cell wall macromolecule catabolic process, amino sugar catabolic process, glucosamine-containing compound metabolic process and glucosamine-containing compound catabolic process. Figure 2 shows that there were significant differences in the transcriptome profiles between non-transgenic lines '717' and '84 K', and between non-transgenic lines (CK) and transgenic plants. In other words, after the transgene was conducted, it was observed that the transcriptional group of upregulated and downregulated genes had undergone great changes.

## 3.4. Changes on pathways in transgenic seedlings

Multiple branches were found in the *ARK1* transgenic seedlings. Zeatin and brassinosteroids were found to be related to cell division or shoot initiation [27,28], and phenylpropanoid was found to be related to abnormal growth of plant phenotype [10]. Hence, we inferred that the abnormal growth of the *ARK1* transgenic plants is related to gene expression levels changing in the pathways of zeatin and brassinosteroids signalling and phenylpropanoid biosynthesis. In the zeatin signalling pathway of transgenic poplars '717' and '84 K', the expression levels of the *AHP* gene was upregulated and the expression of other genes remained unchanged (figure 3*a*); in the pathway of brassinosteroid signalling pathway, the expression of the *CYCD3* gene was upregulated and the expression of other genes was unchanged (figure 3*b*). In the pathway of phenylpropanoid biosynthesis, the expression of *PER27* [EC:1.11.1.7] of transgenic '717' and '84 K' was downregulated, while the expression of β-glucosidase [EC:3.2.1.21] was upregulated, compared with the control plants (figure 3*c*). However, the expression of trans-cinnamate 4-monooxygenase [EC:1.14.13.11] was downregulated in transgenic '717', and the rest of the genes did not change. The expression of 4-coumarate-CoA ligase [EC:6.2.1.12] in '84 K' poplar was upregulated, while that of shikimate O-(hydroxycinnamoyl)transferase [EC:2.3.1.133] was downregulated, and the expression of other genes did not change (figure 3*d*). It indicated that in different varieties, after the introduction of the *ARK1* gene, the gene expression of the same gene regulatory network could also be different.

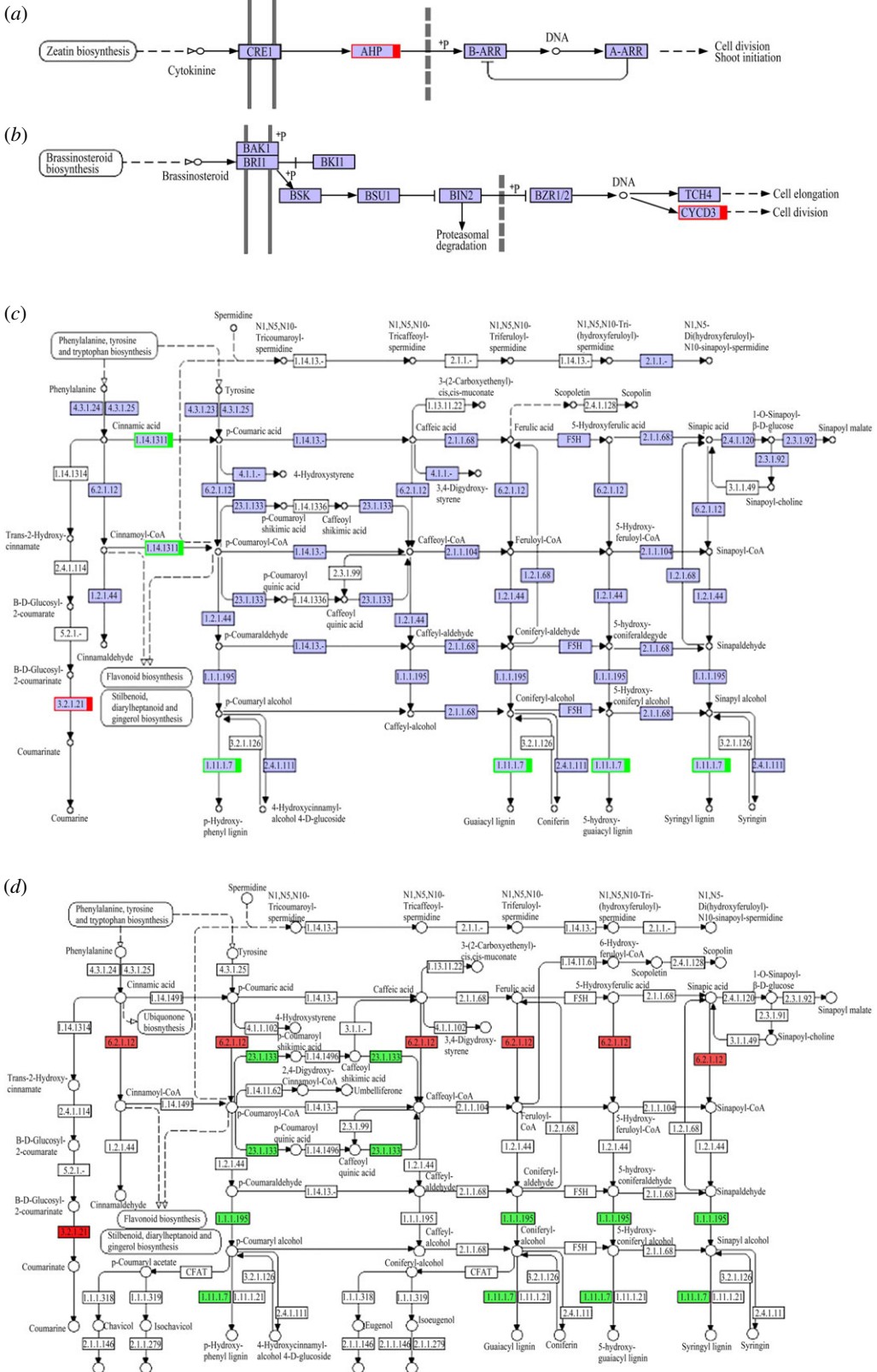

**Figure 3.** Gene expression level changes in the pathway. (*a*) Zeatin biosynthesis pathway; (*b*) brassinosteroid biosynthesis pathway; (*c*) phenylpropanoid biosynthesis pathway of '717'; (*d*) phenylpropanoid biosynthesis pathway of '84 K'; red, upregulated; green, downregulated. *AHP* upregulated 2.71 times and 2.46 times (*p*-value = 0.00052 and 0.00122, significant at *p* < 0.01) in transgenic '717' and '84 K' lines comparing with control plants; *CYCD3* upregulated 1.26 times and 1.48 times (*p*-value = 0.0076 and 0.00219, significant at *p* < 0.01) in transgenic lines, respectively; *PER27* downregulated 1.56 times and 1.82 times (*p*-value = 0.00093 and 0.00133, significant at *p* < 0.01), respectively. (*a,b,c,d*) were KEGG pathway maps [29].

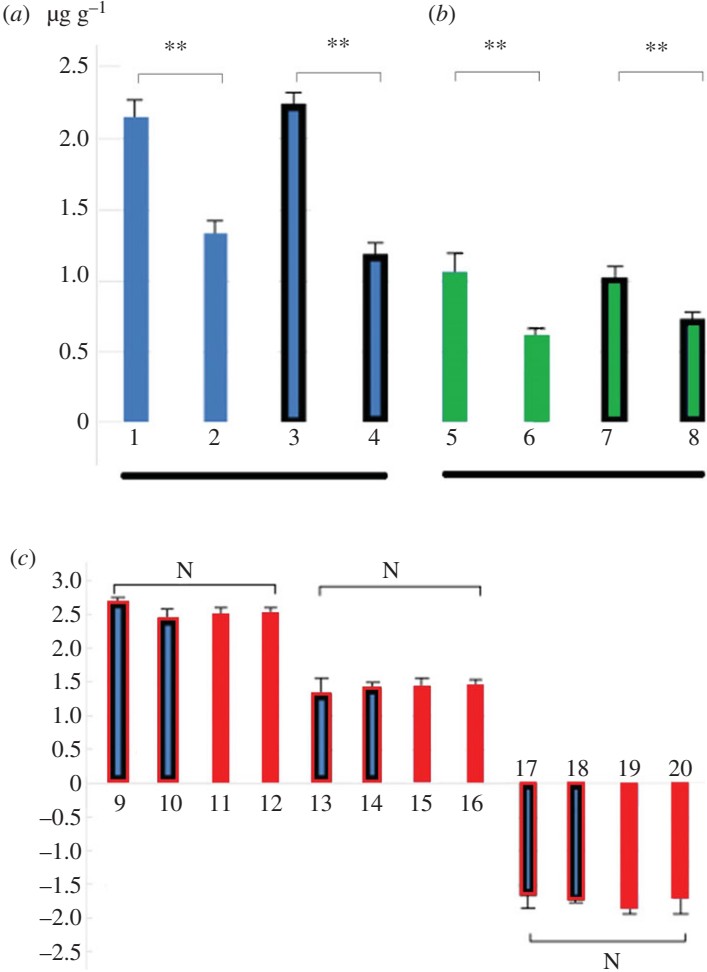

**Figure 4.** Stem hormone content and RT-qPCR verification of transgenic seedlings and non-transgenic seedlings. (*a*) Content of zeatin; (*b*) content of brassinosteroid; (*c*) comparison of RNA-sequencing and RT-qPCR results of selected DEGs. 1, 5, lines of transgenic '717'; 3, 7, lines of transgenic '84 K'; 2, 6, none-transgenic '717'; 4, 8, none-transgenic '84 K'; **, the difference is significant at *p*-value < 0.01; N, the difference was not significant. The samples were harvested after 45 days of culture, RNA-seq, RT-qPCR. 9–12, *AHP*; 13–16, *CYCD3*; 17–20, *PER27*. 9,13,17, lines of transgenic '717'; 10,14,18, lines of transgenic '84 K'; 11,15,19, none-transgenic '717'; 12,16,20, none-transgenic '84 K'. RT-qPCR was performed on three transgenic and three control lines, normalized with housekeeping gene elongation factor *EF1β*, repeated three times.

## 3.5. Determination of hormone content and RT-qPCR verification of gene expression

The contents of zeatin and brassinosteroids in the stems of transgenic plants were significantly higher than those of non-transgenic plants (figure 4*a*,*b*). There was no significant difference in the content of these two hormones between transgenic plants and non-transgenic plants. There was no significant difference between the results of the gene expression of *AHP*, *CYCD3* and *PER27* revealed by RT-qPCR analysis and the results of transcriptome analysis (figure 4*c*). It confirmed that the gene expression levels of *AHP* and *CYCD3* were upregulated, and the expression levels of *PER27* were downregulated in the *ARK1* transgenic plants, comparing with the non-transgenic plants.

# 4. Discussion

Groover *et al.* [6] cloned the *ARK1*, a homologous gene of *Arabidopsis* SHOOT MERISTEMLESS (*STM*), in poplar for the first time. It was found that *ARK1* was mainly expressed in the cambium. Through the analysis of *ARK1* transgenic materials, it was found that *ARK1* and *STM* were expressed in the apical meristem and vascular cambium, but downregulated in the terminal cells of leaf and vascular tissue differentiation. It is speculated that *ARK1* may regulate cell fate through extracellular matrix modification [6]. Liu *et al.* [7] used chromatin co-immunoprecipitation sequencing (chip-seq) to

determine the binding sites of *ARK1* in the poplar genome and found that *ARK1* binds to thousands of sites, which are highly enriched in the transcriptional initiation sites of genes with different functions. Through the experiment, it was found that only 6.2% of the target genes were differentially expressed in the *ARK1* overexpression mutants and 73.2% of the differentially expressed target genes were upregulated, indicating that the abnormality of *ARK1* had little effect on the target genes and *ARK1* was related to transcriptional activation [7]. Ye at al. [22] found that overexpression of *ARK1* gene in hybrid poplar '717' led to downregulation of lignin synthesis-related genes. In this study, the *ARK1* gene of '84 K' poplar was constructed into the transgenic vector, and the transgenic plants of '717' and '84 K' poplar with the *ARK1* gene were obtained. There were differences in seedling branches of transgenic hybrid poplars in this study, and the results were consistent with the results of previous studies [6].

There are lots of genes either up- or downregulated in the transgenic lines from RNA-seq data. Although any gene is overexpressed in a plant, it would cause transcriptomic changes. The expression levels of specific genes changed is unique to *ARK1* overexpressed. Through the analysis of the DEGs in zeatin and brassinosteroid synthesis pathways, most of the related genes in these two synthesis pathways were upregulated, which means that the growth-related genes were upregulated in '717' and '84 K' hybrid poplars transformed with *ARK1*, indicating that the *ARK1* gene had a positive regulatory effect on plant growth. The expression of lignin synthesis-related enzymes in the phenylpropane pathway is downregulated (as shown in figure 1), while the decrease of lignin synthesis leads to abnormal growth of plant phenotype in height, diameter and so on [10]. Zeatin is related to cell division and shoots initiation, and brassinosteroids are related to cell division [27,28]. The results of this study showed that the content of zeatin and brassinosteroids in the stem of transgenic plants with the expression levels of the *ARK1* gene was higher than that of non-transgenic plants and showed a state of multi-branching and active buds in the stems in the phenotype (figure 1).

Oka *et al.* [30] found that the *Arabidopsis thaliana* has five genes named *AHP1* to *AHP5*, and *AHP1* to *AHP5*, each encoded with an HPt factor, are involved in the pathway response to cytokinins. Tanaka *et al.* [31] found that the *HAP* is involved in the signal transduction pathway in response to cytokinin. In our study, the results showed that overexpression of the *ARK1* gene leads to the upregulation of *AHP* and the increase of zeatin contents. Wang *et al.* [32] cloned the *CYCD3-1* gene from the *A. thaliana* genome and inserted it into a plant binary vector pER8, which was controlled by a chimeric transcriptional promoter. A foreign gene was introduced into *A. thaliana* by the *Agrobacterium tumefaciens*-mediated vacuum infiltration method. The results showed that the low-level misexpression of *CYCD3* would affect the growth and development of plants [32]. In our study, the expression of *CYCD3* in transgenic plants was significantly higher than that in non-transgenic plants. Its proteolysis was found to be proteasome-dependent, and the expression levels are dependent on the protein synthesis rate [33]. *CYCD3* encoding a highly unstable cyclin D-type protein is transcriptionally regulated by cytokinin and brassinosteroids and plays a role in cell proliferation and differentiation [33]. It interacts with *ICK1*, which is a cyclin-dependent kinase inhibitor. Overexpressing *CYCD3* in plants could lead to extensive leaf curling, increased leaf number and disorganized meristems [33]. This phenomenon is also reflected in the leaves of our transgenic poplars. Our study showed that overexpression of the *ARK1* gene leads to the upregulation of *CYCD3*, which leads to leaf curling too.

## 5. Conclusion

Overexpression of the *ARK1* gene can upregulate the expression of the *AHP* gene in the zeatin signalling pathway of the plant stem and *CYCD3* gene of the brassinosteroid signalling pathway and leads to the increase of plant branching and the changes of transcriptome profiles. The *ARK1* gene is involved in the growth process of poplar.

Abbreviations. 6-BA: 6-benzyl aminopurine; BIG: Beijing Institute of Genomics; CTAB: cetyl trimethylammonium bromide; DEGs: differentially expressed genes; GO: gene ontology; GOID: gene ontology identity document; IBA: indole-3-butytric acid; MS: Murashige and Skoog; NAA: naphthalene acetic acid; PCR: polymerase chain reaction; RT-qPCR: real-time quantitative polymerase chain reaction; ZT: zeatin.
Ethics. The experiment materials do not include human being or animal. Hence, ethics approval and consent to participate is not applicable.
Data accessibility. All data generated or analysed during this study are included in this published article. RNA-Seq data were presented at the NCBI Sequence Read Archive (accession no. PRJNA633952). The RNA-Seq data of hybrid poplar '717' has been published in a previous study [22].
Authors' contributions. H.Z. and A.D. conceived and designed the experiments. X.L., Z.Z. and W.B. performed the experiments. X.L. and Z.Z. analysed the data, and wrote the paper. Z.Z. and W.B. contributed analysis tools. X.L. edited the paper. All authors have read and approved the manuscript.

Competing interests. The authors declare that they have no competing interests.

Funding. The project was support from the National key R & D Plan for the 13th Five-Year Plan Project of China (grant no. 2016YFD0600102) and the National Natural Science Foundation of China (grant no. 31760450). The funders had no role in the design of the study and collection, analysis and interpretation of data and in writing the manuscript.

Acknowledgements. The manuscript was proofread by Proofed Inc. (UK).

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
