## [Reviewer comments · Royal Society Open Science]

Review History

RSOS-200568.R0 (Original submission)

Review form: Reviewer 1

Is the manuscript scientifically sound in its present form?

Yes

Are the interpretations and conclusions justified by the results?

Yes

Is the language acceptable?

Yes

Do you have any ethical concerns with this paper?

No

Have you any concerns about statistical analyses in this paper?

No

Recommendation?

Major revision is needed (please make suggestions in comments)

Comments to the Author(s)

Major concerns:

1. The plasmid map should be provided in the section of M&M.
2. The confirmation of successfully transgenic events is not satisfied. It needs at least additional Southern blotting not just using PCR only.
3. There is a lack of a control plant (non-transgenic) in Figure 1 for comparing.
4. Figure 1 is out of focus, and the resolution of all figures is not satisfying.

Grammar check:

Line54: The KNOX gene family encodes homeobox proteins, which plays an important regulatory role - The KNOX gene family encodes homeobox proteins, which play an important regulatory role

Line57: By regulating cell differentiation related to organogenesis - By regulating cell differentiation that related to organogenesis

Review form: Reviewer 2

Is the manuscript scientifically sound in its present form?

No

Are the interpretations and conclusions justified by the results?

No

Is the language acceptable?

No

Do you have any ethical concerns with this paper?

Yes

Have you any concerns about statistical analyses in this paper?

No

Recommendation?

Reject

Comments to the Author(s)

In the manuscript submitted by Lu et al., the authors characterized the phenotypic and transcriptomic changes in the ARK1 overexpression lines compared to wild type Populus lines. There are some important issues that need to be addressed:

Outstanding Comment:

1. The material of "717" transgenic line and the transcriptome data of "717" transgenic line are from the published paper "Over-expression of transcription factor ARK1 gene leads to downregulation of lignin synthesis related genes in hybrid poplar "717"" (Ye et al. Scientific Report, 2020), which is from the same lab of this current manuscript. However, in this manuscript, we could not see any reference or acknowledgement. We are not sure how the publication of these two very related manuscripts is coordinated.

Major Comments:

2. Result 3.1: to declare that the transgenic lines have the phenotype resulted from over expressed ARK1 gene, qPCR data of ARK1 is the most reliable method. The authors also need to

show the future readers that what is the promoter for enhancing ARK1 gene? How many transgenic lines did you get? How strong the did the ARK1 express in transgenic lines? How many lines did choose to maintain from qPCR data? What are the reference genes? And so on. The authors declared in the manuscript title that overexpressed ARK1 leads to multiple branches in poplar. But the authors did not document branch numbers. There is no evidence to support the statement. Moreover, in Figure 1, there is no wild type control, no scale bar, and the future readers will not understand how old these seedlings are.

3. Result 3.2: This part has transcriptomic analysis from transgenic "717" and transgenic "84K". The data, for instance, DEG numbers between transgenic "717" and WT "717" are different from it in another paper (Ye et al. Scientific Report, 2020) published by the authors' lab. The authors did not explain the reason.

The Figure 2 could be more informative, for instance, describe what genes are up-regulated and what are down-regulated.

4. Result 3.3: There are thousands of DEGs. It is not clear for us why did the authors choose zeatin, brassinosteroids, phenylpropanoid pathways to give a deeper transcriptomic description? Are they relevant to the branching phenotype? If so, the authors should give evidence or some hypothesis that how those genes would affect those pathways and further affect the phenotype. There are some mistakes, for instance, AHP and CYCD3 they are not in biosynthesis pathways, but in signaling pathways.

Figure 3: how did the authors get the maps? From KEGG?

The authors marked some up- and down-regulated genes on the maps, but there is not transcriptomic data, which could support the Figure 3. The author should specifically show those genes changed expression level from result 3.2 data.

5. Result 3.4: The data in this part is quite controversial. AHP and CYCD3 are up-regulated according to result 3.3. Why they are not up-regulated in result 3.4, according to Figure 4C? And why the zeatin and brassinosteroids profiles changes in Figure 4A?

Figure 4 is quite difficult to read. We recommend the authors re-organize the data in the plots, with clear plot legends and figure legends. This also applies to other figures in this manuscript.

6. The manuscript is not well written enough scientifically and grammarly. The authors did not introduce zeatin, brassinosteroids, phenylpropanoid pathways, either branching studies. And in the discussion, some parts are not very relevant to this manuscript. Lacking reference for some statements is also an issue in this manuscript. Figures are not clear to read. We encourage the authors to re-shape the manuscript and analyze the data more thoroughly and carefully.

Decision letter (RSOS-200568.R0)

Dear Dr Zhang:

Manuscript ID RSOS-200568 entitled "Enhancing the expression of ARK1 genes in poplar leads to multiple branches and transcriptome profile changes" which you submitted to Royal Society Open Science, has been reviewed. The comments from reviewers are included at the bottom of this letter.

In view of the criticisms of the reviewers, the manuscript has been rejected in its current form. Both reviewers raise substantive issues with the data and its analysis, and neither reviewer believes the manuscript is publishable in its current form. However, a new manuscript may be submitted which takes into consideration these comments.

Please note that resubmitting your manuscript does not guarantee eventual acceptance, and that your resubmission will be subject to peer review before a decision is made.

Your resubmitted manuscript should be submitted by 06-Dec-2020. If you are unable to submit by this date please contact the Editorial Office.

on behalf of Dr James Locke (Associate Editor) and Steve Brown (Subject Editor)
openscience@royalsociety.org

Associate Editor Comments to Author (Dr James Locke):

Comments to the Author:

Although the work is of interest, the two reviewers both point out major problems with the data and its interpretation. Unfortunately this means we must reject the manuscript.

Reviewers' Comments to Author:

Reviewer: 1

Comments to the Author(s)

Major concerns:

1. The plasmid map should be provided in the section of M&M.
2. The confirmation of successfully transgenic events is not satisfied. It needs at least additional Southern blotting not just using PCR only.
3. There is a lack of a control plant (non-transgenic) in Figure 1 for comparing.
4. Figure 1 is out of focus, and the resolution of all figures is not satisfying.

Grammar check:

Line54: The KNOX gene family encodes homeobox proteins, which plays an important regulatory role - The KNOX gene family encodes homeobox proteins, which play an important regulatory role

Line57: By regulating cell differentiation related to organogenesis - By regulating cell differentiation that related to organogenesis

Reviewer: 2

Comments to the Author(s)

In the manuscript submitted by Lu et al., the authors characterized the phenotypic and transcriptomic changes in the ARK1 overexpression lines compared to wild type Populus lines. There are some important issues that need to be addressed:

Outstanding Comment:

1. The material of "717" transgenic line and the transcriptome data of "717" transgenic line are from the published paper "Over-expression of transcription factor ARK1 gene leads to downregulation of lignin synthesis related genes in hybrid poplar '717'" (Ye et al. Scientific Report, 2020), which is from the same lab of this current manuscript. However, in this manuscript, we could not see any reference or acknowledgement. We are not sure how the publication of these two very related manuscripts is coordinated.

Major Comments:

2. Result 3.1: to declare that the transgenic lines have the phenotype resulted from over expressed ARK1 gene, qPCR data of ARK1 is the most reliable method. The authors also need to show the future readers that what is the promoter for enhancing ARK1 gene? How many transgenic lines did you get? How strong the did the ARK1 express in transgenic lines? How many lines did choose to maintain from qPCR data? What are the reference genes? And so on.

The authors declared in the manuscript title that overexpressed ARK1 leads to multiple branches in poplar. But the authors did not document branch numbers. There is no evidence to support the statement. Moreover, in Figure 1, there is no wild type control, no scale bar, and the future readers will not understand how old these seedlings are.

3. Result 3.2: This part has transcriptomic analysis from transgenic "717" and transgenic "84K". The data, for instance, DEG numbers between transgenic "717" and WT "717" are different from it in another paper (Ye et al. Scientific Report, 2020) published by the authors' lab. The authors did not explain the reason.

The Figure 2 could be more informative, for instance, describe what genes are up-regulated and what are down-regulated.

4. Result 3.3: There are thousands of DEGs. It is not clear for us why did the authors choose zeatin, brassinosteroids, phenylpropanoid pathways to give a deeper transcriptomic description? Are they relevant to the branching phenotype? If so, the authors should give evidence or some hypothesis that how those genes would affect those pathways and further affect the phenotype. There are some mistakes, for instance, AHP and CYCD3 they are not in biosynthesis pathways, but in signaling pathways.

Figure 3: how did the authors get the maps? From KEGG?

The authors marked some up- and down-regulated genes on the maps, but there is not transcriptomic data, which could support the Figure 3. The author should specifically show those genes changed expression level from result 3.2 data.

5. Result 3.4: The data in this part is quite controversial. AHP and CYCD3 are up-regulated according to result 3.3. Why they are not up-regulated in result 3.4, according to Figure 4C? And why the zeatin and brassinosteroids profiles changes in Figure 4A?

Figure 4 is quite difficult to read. We recommend the authors re-organize the data in the plots, with clear plot legends and figure legends. This also applies to other figures in this manuscript.

6. The manuscript is not well written enough scientifically and grammatically. The authors did not introduce zeatin, brassinosteroids, phenylpropanoid pathways, either branching studies. And in the discussion, some parts are not very relevant to this manuscript. Lacking reference for some statements is also an issue in this manuscript. Figures are not clear to read. We encourage the authors to re-shape the manuscript and analyze the data more thoroughly and carefully.

Author's Response to Decision Letter for (RSOS-200568.R0)

See Appendix A.

RSOS-201201.R0

Review form: Reviewer 1

Is the manuscript scientifically sound in its present form?

Yes

Are the interpretations and conclusions justified by the results?

Yes

Is the language acceptable?

Yes

Do you have any ethical concerns with this paper?

No

Have you any concerns about statistical analyses in this paper?

No

Recommendation?

Accept as is

Comments to the Author(s)

The manuscript has been improved accordingly, and I don't have any further questions.

Review form: Reviewer 2

Is the manuscript scientifically sound in its present form?

No

Are the interpretations and conclusions justified by the results?

No

Is the language acceptable?

Yes

Do you have any ethical concerns with this paper?

No

Have you any concerns about statistical analyses in this paper?

No

Recommendation?

Reject

Comments to the Author(s)

In the manuscript resubmitted by Lu et al., the authors characterized the phenotypic and transcriptomic changes in the ARK1 overexpression lines compared to wild type Populus lines. The authors improved this manuscript, but there are still some important issues that need to be addressed:

Major Comments:

1. The quality of the current manuscript is still not good enough. Even though, probably the authors tried to make a small story, it is need to be a qualified story. We do not get why the authors decided to study this ARK1 gene. The authors said that "The ARBORKNOX1 (ARK1) gene is an important gene for regulating plant growth and development; however, its gene network regulation is poorly investigated." There are many genes are important for plant growth. Moreover, the authors did not really studied the network regulation. There are genes either up- or down-regulated in the transgenic lines from RNA-seq data. However, any (if not, most genes should be the case) gene is over-expressed in plant, would cause transcriptomic changes. The authors did not study the hierarchical relationship between ARK1 and transcription factors, associated proteins, and their target genes. There are transcriptomic changes in the transgenic plants. But, how does ARK1 affect them?
2. There are thousands of DEGs in the ARK1 overexpressed lines and there are many pathways or hormones affect branching phenotype in nature. It is still not clear for us why did the authors choose zeatin, brassinosteroids, phenylpropanoid pathways to give a deeper transcriptomic description. The authors listed out some pathways relevant to branching morphology, including stigolactone pathway in the Introduction Section. However, the conclusion was "the researches on zeatin and brassinosteroids signaling pathways and phenylpropanoid biosynthesis pathways should be focused on". Why not strigolactone? This is one example that this manuscript is not scientifically and logically sufficient. There are more.
3. All the figures are not qualified enough for publication. Need to be made carefully and readable.

Minor Comments:

1. Result 3.1: the authors declared that the ARK1 gene is up-regulated in transgenic lines about 3 times compare to control lines and up-regulated lines had branches about 5 times more control lines. However, there is quantification figures or raw data.
2. It is good the authors provide plasmid map and performed southern blot as another reviewer suggested. However, these two figures should be in supplementary figures.

Decision letter (RSOS-201201.R0)

Dear Dr Zhang,

On behalf of the Editor, I am pleased to inform you that your Manuscript RSOS-201201 entitled "Enhancing the expression of ARK1 genes in poplar leads to multiple branches and transcriptome profile changes" has been accepted for publication in Royal Society Open Science subject to minor revision in accordance with the referee suggestions. Please find the referees' comments at the end of this email.

The reviewers, Associate Editor and Subject Editor have recommended publication, but also suggest some minor revisions to your manuscript. Please note carefully the points of the Associate Editor in responding to the comments of Reviewer 2. It will be important to revise the

manuscript accordingly as indicated by the Associate Editor. Therefore, I invite you to respond to the comments and revise your manuscript.

- Ethics statement

- Data accessibility

<http://datadryad.org/submit?journalID=RSOS&manu=RSOS-201201>

- Competing interests

- Authors' contributions

- Acknowledgements

- Funding statement

Because the schedule for publication is very tight, it is a condition of publication that you submit the revised version of your manuscript before 07-Aug-2020. Please note that the revision deadline will expire at 00.00am on this date. If you do not think you will be able to meet this date please let me know immediately.

on behalf of Dr James Locke (Associate Editor) and Steve Brown (Subject Editor)
openscience@royalsociety.org

Associate Editor Comments to Author (Dr James Locke):

The reviewers appreciate your efforts on the revision. However, one reviewer still has issues with some of the claims and the justifications of the manuscript. I propose that in a further revision you downplay some of your claims as the reviewer suggests and justify further your choice of focus (as well as moving the plasmid map etc into supplementary).

Additionally, the RNAseq data should be made publicly available before publication. There is currently no result after searching PRJNA633952 in NCBI, which is the reference given for the dataset.

Reviewer comments to Author:

Reviewer: 1

Comments to the Author(s)

The manuscript has been improved accordingly, and I don't have any further questions.

Reviewer: 2

Comments to the Author(s)

In the manuscript resubmitted by Lu et al., the authors characterized the phenotypic and transcriptomic changes in the ARK1 overexpression lines compared to wild type Populous lines. The authors improved this manuscript, but there are still some important issues that need to be addressed:

Major Comments:

1. The quality of the current manuscript is still not good enough. Even though, probably the authors tried to make a small story, it is need to be a qualified story. We do not get why the authors decided to study this ARK1 gene. The authors said that "The ARBORKNOX1 (ARK1) gene is an important gene for regulating plant growth and development; however, its gene network regulation is poorly investigated." There are many genes are important for plant growth. Moreover, the authors did not really studied the network regulation. There are genes either up- or down-regulated in the transgenic lines from RNA-seq data. However, any (if not, most genes should be the case) gene is over-expressed in plant, would cause transcriptomic changes. The authors did not study the hierarchical relationship between ARK1 and transcription factors, associated proteins, and their target genes. There are transcriptomic changes in the transgenic plants. But, how does ARK1 affect them?

2. There are thousands of DEGs in the ARK1 overexpressed lines and there are many pathways or hormones affect branching phenotype in nature. It is still not clear for us why did the authors choose zeatin, brassinosteroids, phenylpropanoid pathways to give a deeper transcriptomic description.

The authors listed out some pathways relevant to branching morphology, including stigolactone pathway in the Introduction Section. However, the conclusion was "the researches on zeatin and brassinosteroids signaling pathways and phenylpropanoid biosynthesis pathways should be focused on". Why not strigolactone? This is one example that this manuscript is not scientifically and logically sufficient. There are more.

3. All the figures are not qualified enough for publication. Need to be made carefully and readable.

Minor Comments:

1. Result 3.1: the authors declared that the ARK1 gene is up-regulated in transgenic lines about 3 times compare to control lines and up-regulated lines had branches about 5 times more control lines. However, there is quantification figures or raw data.
2. It is good the authors provide plasmid map and performed southern blot as another reviewer suggested. However, these two figures should be in supplementary figures.

Author's Response to Decision Letter for (RSOS-201201.R0)

See Appendix B.

Decision letter (RSOS-201201.R1)

Dear Dr Zhang,

It is a pleasure to accept your manuscript entitled "Enhancing the expression of ARK1 genes in poplar leads to multiple branches and transcriptomic changes" in its current form for publication in Royal Society Open Science.

on behalf of Dr James Locke (Associate Editor) and Steve Brown (Subject Editor)
openscience@royalsociety.org

Appendix A

Dear Editor:

We have revised the manuscript according to the reviewers comments. Following are the answers to the reviewers concerns:

Reviewers' Comments to Author:

Reviewer: 1

Comments to the Author(s)

Major concerns:

1. The plasmid map should be provided in the section of M&M.

Answer: Provided.

2. The confirmation of successfully transgenic events is not satisfied. It needs at least additional Southern blotting not just using PCR only.

Answer: We have added the experiments of Southern blotting.

3. There is a lack of a control plant (non-transgenic) in Figure 1 for comparing.

Answer: We have added control plants.

4. Figure 1 is out of focus, and the resolution of all figures is not satisfying.

Answer: Corrected.

Grammar check:

Line54: The KNOX gene family encodes homeobox proteins, which plays an important regulatory role -
The KNOX gene family encodes homeobox proteins, which play an important regulatory role

Answer: Corrected.

Line57: By regulating cell differentiation related to organogenesis - By regulating cell differentiation that
related to organogenesis

Answer: Corrected.

Reviewer: 2

Comments to the Author(s)

In the manuscript submitted by Lu et al., the authors characterized the phenotypic and transcriptomic changes in the ARK1 overexpression lines compared to wild type Populous lines. There are some important issues that need to be addressed:

Outstanding Comment:

1. The material of "717" transgenic line and the transcriptome data of "717" transgenic line are from the published paper "Over-expression of transcription factor ARK1 gene leads to downregulation of lignin synthesis related genes in hybrid poplar '717'" (Ye et al. Scientific Report, 2020), which is from the same lab of this current manuscript. However, in this manuscript, we could not see any reference or acknowledgement. We are not sure how the publication of these two very related manuscripts is coordinated.

Answer: We have cited the published paper "Over-expression of transcription factor ARK1 gene leads to down regulation of lignin synthesis related genes in hybrid poplar '717'" (Ye et al. Scientific Report, 2020) in the new version. The published paper mainly described the relationship between ARK1 and lignin synthesis related genes in hybrid poplar '717'. In this paper, we mainly describe the relationship between ARK1 and the multiple branches, and we have included the transcriptome data of '84K' .

Major Comments:

2. Result 3.1: to declare that the transgenic lines have the phenotype resulted from over expressed ARK1 gene, qPCR data of ARK1 is the most reliable method. The authors also need to show the future readers that what is the promotor for enhancing ARK1 gene? Answer: We have used the *CaMV 35S* as the promotor for enhancing ARK1 gene. How many transgenic lines did you get? Answer: Eleven out of 35 '717' seedlings and 14 out of 41 '84K' seedlings with a transformation marker were identified, and a total of 32.22% plants were found to be PCR positive. How strong the did the ARK1 express in transgenic lines? Answer: The results of the RT-qPCR analysis showed that the average expression levels of the *ARK1* gene in transgenic '717' and '84K' lines was about 2.95 times and 3.16 times that of the control seedlings. How many lines did choose to maintain from qPCR data? Answer: We found that the expression levels of eight transgenic '717' lines and ten transgenic '84K' lines were found to be at least twice more than that of the control seedlings. What are the reference genes? And so on. Answer: We used the *EF1 β* as the reference gene.

The authors declared in the manuscript title that overexpressed ARK1 leads to multiple branches in poplar. But the authors did not document branch numbers. There is no evidence to support the statement. Answer: After cultured about 45 days, the branches in all PCR positive '717' and '84K' lines were about 4.73 times and 5.24 times that of control seedlings. Moreover, in Figure 1, there is no wild type control, no scale bar, and the future readers will not understand how old these seedlings are. Answer: We have added all the information needed.

3. Result 3.2: This part has transcriptomic analysis from transgenic "717" and transgenic "84K". The data, for instance, DEG numbers between transgenic "717" and WT "717" are different from it in another paper (Ye et al. Scientific Report, 2020) published by the authors' lab. The authors did not explain the reason. Answer: We used the threshold $\text{abs}(\log_2\text{FoldChange}) > 0$ to screen the DEGs before, so it is different to the data of Ye et al. Scientific Report, 2020, now we use the threshold $\text{abs}(\log_2\text{FoldChange})$

> 1 to screen.

The Figure 2 could be more informative, for instance, describe what genes are up-regulated and what are down-regulated.

Answer: We have described the up-regulated and down-regulated genes.

4. Result 3.3: There are thousands of DEGs. It is not clear for us why did the authors choose zeatin, brassinosteroids, phenylpropanoid pathways to give a deeper transcriptomic description? Are they relevant to the branching phenotype? If so, the authors should give evidence or some hypothesis that how those genes would affect those pathways and further affect the phenotype. Answer: Zeatin and brassinosteroids were found to be related to cell division or shoots initiation (Taylor et al., 1984; Lipka et al., 2015), and phenylpropanoid was found to be related to abnormal growth of plant phenotype (Song and Wang, 2011). Hence, Zeatin, brassinosteroids, phenylpropanoid pathways were chose to give a deeper transcriptomic description. There are some mistakes, for instance, AHP and CYCD3 they are not in biosynthesis pathways, but in signaling pathways. Answer: Corrected.

Figure 3: how did the authors get the maps? From KEGG?

Answer: From KEGG, and we have cited the reference.

The authors marked some up- and down-regulated genes on the maps, but there is not transcriptomic data, which could support the Figure 3. The author should specifically show those genes changed expression level from result 3.2 data.

Answer: We have revised it according to the reviewers' comments.

5. Result 3.4: The data in this part is quite controversial. AHP and CYCD3 are up-regulated according to result 3.3. Why they are not up-regulated in result 3.4, according to Figure 4C? And why the zeatin and brassinosteroids profiles changes in Figure 4A?

Answer: I think that I have expressed the results in the wrong way to let the reviewers misunderstand. The gene expression of AHP, CYCD3, and PER27 revealed by RT-qPCR analysis was basically the same as that of transcriptome analysis, and there was no significant difference (see Figure 6C).

Figure 4 is quite difficult to read. We recommend the authors re-organize the data in the plots, with clear plot legends and figure legends. This also applies to other figures in this manuscript.

Answer: We have revised it according to the reviewers' comments.

6. The manuscript is not well written enough scientifically and grammarly. The authors did not introduce zeatin, brassinosteroids, phenylpropanoid pathways, either branching studies. Answer: We have discussed the relationships between the branching and zeatin, brassinosteroids and phenylpropanoid pathways in the Discussion part, although we did not present them in the Introduction part. Because we

found that zeatin, brassinosteroids and phenylpropanoid pathways are related to branching after we analysis the transcriptome data, we thought it is not proper to present it in the Introduction part. And in the discussion, some parts are not very relevant to this manuscript. Answer: **We have deleted the parts that is not very relevant to this manuscript.** “~~The main synthetic site of zeatin and brassinosteroids is in the root, and they will enter the stem and leaf with the transportation of water. When the content in the stem increases, it will lead to the active bud in the stem.~~” Lacking reference for some statements is also an issue in this manuscript. Answer: **We have added some references in the manuscript.** Figures are not clear to read. Answer: **We have improved the resolution of the figures.** We encourage the authors to re-shape the manuscript and analyze the data more thoroughly and carefully. Answer: **We have re-shaped and revised the manuscript thoroughly and carefully.**

Thanks for your help.

Cheers,

Yours sincerely,

Hanyao Zhang

Professor of Molecular Genetics

Southwest Forestry University

Appendix B

Dear editor,

Many thanks for the letter. We have revised the manuscript according to the editor and reviewer's comment. Because we are not English native speaker, there are may be some places that we did not express very well. If there any thing we do in the wrong way, please let us know, we will try our best to revise it.

Associate Editor Comments to Author (Dr James Locke):

The reviewers appreciate your efforts on the revision. However, one reviewer still has issues with some of the claims and the justifications of the manuscript. I propose that in a further revision you downplay some of your claims as the reviewer suggests and justify further your choice of focus (as well as moving the plasmid map etc into supplementary). Answer: Thanks. We have revised the manuscript according to the reviewer's comment. We have try to downplay some claims. For example, we have deleted the sentence "ARK1 is an important growth-related gene" from the conclusion section, and we have changed "its gene network regulation is poorly investigated." into "transcriptomic responses of enhancing-expression of ARK1 gene in poplar are poorly investigated." Also, we have move the Figure 1 and Figure 2 into supplementary.

Additionally, the RNAseq data should be made publicly available before publication. There is currently no result after searching PRJNA633952 in NCBI, which is the reference given for the dataset. Answer: We have made the RNAseq data be publicly available.

Reviewer comments to Author:

Reviewer: 1

Comments to the Author(s)

The manuscript has been improved accordingly, and I don't have any further questions.

Reviewer: 2

Comments to the Author(s)

In the manuscript resubmitted by Lu et al., the authors characterized the phenotypic and transcriptomic changes in the ARK1 overexpression lines compared to wild type Populus lines. The authors improved this manuscript, but there are still some important issues that need to be addressed:

Major Comments:

1. The quality of the current manuscript is still not good enough. Even though, probably the authors tried to make a small story, it is need to be a qualified story. We do not get why the authors decided to study this ARK1 gene. The authors said that "The ARBORKNOX1 (ARK1) gene is an important gene for regulating plant growth and development; however, its gene network regulation is poorly investigated." There are many genes are important for plant growth. Moreover, the authors did not really studied the network regulation. Answer: The growth of plant especial those wood plant is still very slow and is difficult to meet the needs of industrial development for wood, biomass, paper, fuel and biomaterials (Harfouche et al., 2012; Zong et al., 2019). The ARK1 gene has found to be an important gene for regulating plant growth and development in poplar in previous studies (Groover et al., 2006; Liu et al., 2014). To investigate the function of the ARK1 gene may benefit the plant growth to meet the needs. In this study, to analyze the function of the ARK1 gene, the poplar lines '717' and '84K' were subjected to gene transform,

the morphology and transcriptome profiles were compared between transgenic lines and control plants, and the expression levels of some genes were verified by RT-qPCR. The reviewer said that the authors did not really study the network regulation. So we changed "its gene network regulation is poorly investigated." into "transcriptomic responses of enhancing-expression of *ARK1* gene in poplar are poorly investigated." There are genes either up- or down-regulated in the transgenic lines from RNA-seq data.

However, any (if not, most genes should be the case) gene is over-expressed in plant, would cause transcriptomic changes.

Answer: Although any gene is over-expressed in plant, would cause transcriptomic changes. The expression levels of specific genes changed is unique to *ARK1* over-expressed.

The authors did not study the hierarchical relationship between *ARK1* and transcription factors, associated proteins, and their target genes. There are transcriptomic changes in the transgenic plants. But,

how does *ARK1* affect them? Answer: Liu et al. (2014) used chromatin co-immunoprecipitation sequencing (chip-seq) to determine the binding sites of *ARK1* in the poplar genome and found that *ARK1* binds to thousands of sites, which are highly enriched in the transcriptional initiation sites of genes with different functions.

We thought that Liu et al. (2014) should have described the target genes in their study. We are sorry that we don't know how to study the hierarchical relationship between *ARK1* and transcription factors, associated proteins, and their target genes.

Maybe using the chromatin co-immunoprecipitation sequencing we could do more research on the relationships between *ARK1* and other transcription factors, associated proteins, and their target genes, and on how *ARK1* affect the transcriptomic changes.

We did not describe how does *ARK1* affect the transcriptomic changes in the manuscript. Meanwhile, we have described the transcriptomic changes in the transgenic plants.

2. There are thousands of DEGs in the *ARK1* overexpressed lines and there are many pathways or hormones affect branching phenotype in nature. It is still not clear for us why did the authors choose zeatin, brassinosteroids, phenylpropanoid pathways to give a deeper transcriptomic description.

Answer: The stem segment of transgenic poplar '717' and '84K' had multiple branches. Zeatin and brassinosteroids were known to be involved in the branching phenotype, and phenylpropanoid pathway was found to be related to plant development, in our knowledge. Hence, it could be possible to find the reason why the *ARK1*-transgenic plant have multiple branches.

Although there are thousands of DEGs in the *ARK1* overexpressed lines and there are many pathways or hormones affect branching phenotype in nature. We did not find DEGs with $\text{padj} < 0.05$ and $\text{abs}(\log_2\text{FoldChange}) > 1$ in other pathways or hormones affect branching phenotype. Most important, we did not find any other DEGs annotated in Gene Ontology which had been previously reported to be involved in the branching phenotype.

The authors listed out some pathways relevant to branching morphology, including strigolactone pathway in the Introduction Section. However, the conclusion was "the researches on zeatin and brassinosteroids signaling pathways and phenylpropanoid biosynthesis pathways should be focused on". Why not strigolactone? This is one example that this manuscript is not scientifically and logically sufficient. There are more.

Answer: We have checked the gene expression levels involved in the strigolactone pathway, but not DEGs with $\text{padj} < 0.05$ and $\text{abs}(\log_2\text{FoldChange}) > 1$ were found. In addition, we have added or deleted some information to let the paper to be more scientific and logical.

3. All the figures are not qualified enough for publication. Need to be made carefully and readable.

Answer: We have changed the figures into EPS format, and the pixel of some figures was increased from 300 ppi to 500 ppi.

Minor Comments:

1. Result 3.1: the authors declared that the ARK1 gene is up-regulated in transgenic lines about 3 times compare to control lines and up-regulated lines had branches about 5 times more control lines. However, there is quantification figures or raw data.

Answer: Figures were added into supplementary material.

2. It is good the authors provide plasmid map and performed southern blot as another reviewer suggested. However, these two figures should be in supplementary figures.

Answer: The Figures of plasmid map and southern blot had been moved into supplementary.

Many thanks.

Best regards,

Hanyao Zhang

Prof./Dr.

Southwest Forestry University